# Stöber method to amorphous metal-organic frameworks and coordination polymers

Wei Zhang [1,2] ✉, Yanchen Liu[1], Henrik S. Jeppesen [3] & Nicola Pinna [1] ✉

The Stöber method is a widely-used sol-gel route for synthesizing amorphous $SiO_2$ colloids and conformal coatings. However, the material systems compatible with this method are still limited. Herein, we have extended the approach to metal-organic frameworks (MOFs) and coordination polymers (CPs) by mimicking the Stöber method. We introduce a general synthesis route to amorphous MOFs or CPs by making use of a base-vapor diffusion method, which allows to precisely control the growth kinetics. Twenty-four different amorphous CPs colloids were successfully synthesized by selecting 12 metal ions and 17 organic ligands. Moreover, by introducing functional nanoparticles (NPs), a conformal amorphous MOFs coating with controllable thickness can be grown on NPs to form core-shell colloids. The versatility of this amorphous coating technology was demonstrated by synthesizing over 100 core-shell composites from 20 amorphous CPs shells and over 30 different NPs. Besides, various multifunctional nanostructures, such as conformal yolk-amorphous MOF shell, core@metal oxides, and core@carbon, can be obtained through one-step transformation of the core@amorphous MOFs. This work significantly enriches the Stöber method and introduces a platform, enabling the systematic design of colloids exhibiting different level of functionality and complexity.

Colloidal particles have been extensively studied across a range of fields, including chemistry, materials science and condensed matter physics.[1–6] Monodisperse colloids are highly desirable in numerous applications.[5–9] Among these colloids, monodisperse $SiO_2$ spheres, synthesized by Stöber and co-workers, known as the Stöber method, stands out for their stable chemical properties, tunable size, applicability to diverse nanostructures, and straightforward synthesis protocol.[10,11] The Stöber method represented a significant advancement in colloid chemistry and is still a widely-used ammonia-assisted sol-gel reaction to silica-based colloids and coatings.[11] Recently, the Stöber method has been successfully extended to other materials including $TiO_2$[12], resorcinol-formaldehyde (RF)[13] and metal–phenolic coordination spheres[14,15] which further enrich the synthesis approach. However, the material systems compatible with the Stöber method are

still limited, fixed in their chemical compositions and lacking adjustability, which restricts their field of application. Therefore, it is interesting to try to extend the Stöber method to other material systems that are tunable in their chemical compositions and properties.

Coordination polymers (CPs) and their porous subset, metal−organic frameworks (MOFs) are exceptionally versatile in terms of composition and structures.[16–25] Compared to $SiO_2$, the composition, structures, physical and chemical properties of CPs are highly adjustable, because of the tunability of both the organic and inorganic counterparts of the framework. Furthermore, additional functional species can be integrated onto the inner walls of the pores by post-synthesis modifications or into the channels by infiltration.[26,27] Additionally, CPs can be simply transformed into other functional materials through one-step thermal treatment or self-sacrificial template

[1]Department of Chemistry, IRIS Adlershof & The Center for the Science of Materials Berlin, Humboldt-Universität zu Berlin, Brook-Taylor-Str. 2, 12489 Berlin, Germany. [2]Department of Colloid Chemistry, Max Planck Institute of Colloids and Interfaces, 14476 Potsdam, Germany. [3]Deutsches Elektronen-Synchrotron (DESY), Notkestrasse 85, 22607 Hamburg, Germany. ✉e-mail: zhangweq@hu-berlin.de; nicola.pinna@hu-berlin.de

reactions.[28–30] Therefore, extending the Stöber method for the synthesis of amorphous SiO$_2$ spheres to monodisperse amorphous MOFs or CPs colloids is of great interest.

In recent years, many amorphous CPs ranging from colloidal to bulk-glass state have been designed and attracted increasing attention due to some distinctive characteristics compared to crystalline CPs.[31–42] Firstly, the application of crystalline CPs into functional macroscopic systems, such as thin films coatings and bulk materials, is considerably challenged by their inherent brittleness and rigidity. Compared to crystalline CPs, the flexible and disordered nature of amorphous CPs could effectively alleviate these drawbacks and increase the mechanical and chemical robustness.[34] In addition, amorphous CPs have proven to be advantageous in terms of loading of guest molecules and in finely tuning the release characteristics when used in bio-medicine and catalysis.[43–46] The synthesis of nano-/microparticles of amorphous CPs have been regarded as the effective strategy to enhance their performance.[47,48] While some pioneering studies have reported the synthesis of amorphous CPs sphere,[3,49] the synthetic methods are often tailored to specific systems, and more universally applicable strategy are still required. Furthermore, compared to crystalline CPs, less attention has been given to the synthesis of hierarchical amorphous composites. Therefore, the development of an effective a general approach for the synthesis of amorphous CPs is an important addition to the field of CPs and is also expected to demonstrate more applications in the fields of catalysis, drug release, and electronics.

Herein, we present a simple strategy mimicking the Stöber method to synthesize diverse amorphous MOFs (aMOFs) and amorphous CPs (aCPs) spheres and conformal coatings. Similar to amorphous SiO$_2$, the crystallization processes of many CPs are predominantly governed by kinetics. Therefore, by employing a base vapor diffusion method as a precise kinetic control strategy in CPs systems, the sol-gel synthesis of monodisperse amorphous CPs colloids can be achieved. This strategy inherits the advantages of the Stöber method by using a base to finely tune the kinetics of the sol-gel reaction (Fig. 1a). Briefly, triethylamine (TEA) vapor is utilized to control the deprotonation of organic ligands, consequently governing the kinetics of complexation between metal ions and deprotonated ligands. (Fig. 1b). During this sol-gel process, the controlled and slow complexation of metal ions with the progressively "activated" ligands results in the formation of well-shaped amorphous CPs spheres. More importantly, the slowly controllable growth process allows for heterogeneous deposition of aMOF or aCPs on any substrate, independent of specific surface structures, to form diverse conformal core-shell colloids with uniform shell and controlled thickness at the nano-scale. In this work, we report the synthesis of 24 aMOFs and aCPs and their amorphous coatings onto over 30 types of core-nanoparticles. The successful synthesis of these monodisperse amorphous spheres and >100 different core-shell colloids demonstrate the versatility of our strategy. Furthermore, we demonstrate that the core-amorphous MOFs shell skeletons can be easily transformed into other types of nanostructures, including a range of yolk-amorphous MOF shell, core@metal oxides and core@porous carbon structures. This work greatly extends the Stöber method and provides a promising platform for the systematic design of colloids with different functionalities and complexities.

## Results

### Synthesis and characterizations of amorphous ZIF-7 spheres

Zeolitic imidazolate frameworks (ZIFs) represent a prototypical family of Metal-Organic Frameworks.[50] The structures of ZIFs exhibit zeolite framework topologies, wherein all tetrahedrally coordinates atoms consist of transition metals. The network made of linked CoN$_4$ or ZnN$_4$ tetrahedra in ZIFs resemble SiO$_4$ tetrahedra (Supplementary Fig. 1).[50] ZIF-7, as an exemplary representative of this family of compounds, is considered at first.[50] Its framework is formed by a periodic arrangement of Zn ions with benzimidazole ligands (Fig. 2a). Similar to the Stöber method which utilizes a base for promoting the hydrolysis and condensation of tetraethyl orthosilicate, in our case, TEA is used as a base for the control of the nucleation and growth of ZIF-7. In the first step, ZnCl$_2$ and benzimidazole are dissolved in ethanol to form a clear solution. Without

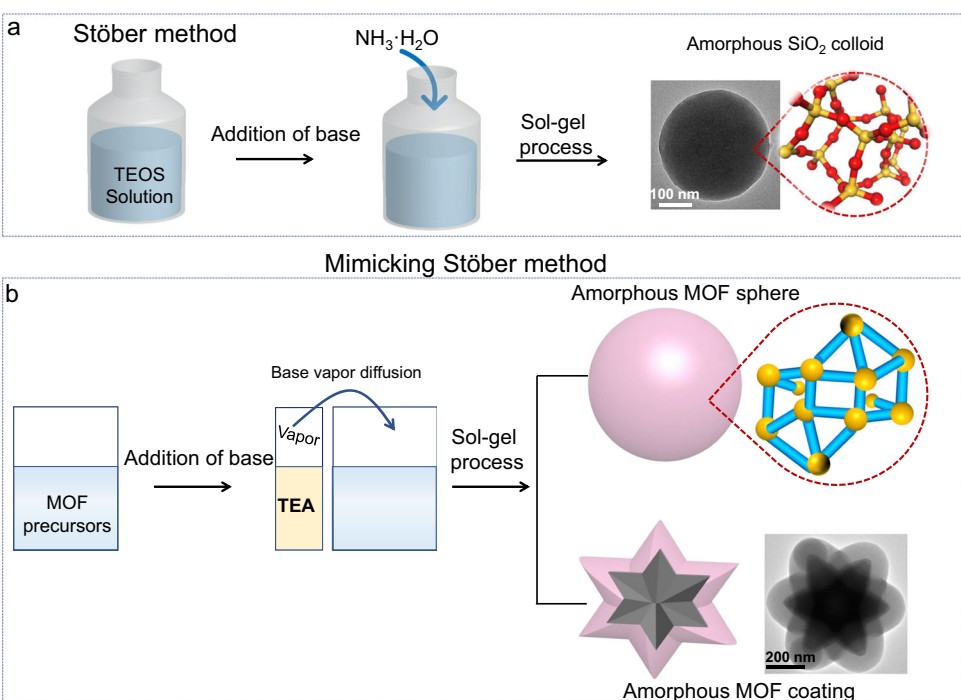

**Fig. 1 | Schematic illustration of the synthesis of aMOFs and aCPs colloids. a** The Stöber method for the synthesis of monodisperse SiO$_2$ spheres (TEOS Tetraethyl orthosilicate). **b** The synthesis of aMOFs and aCPs colloids and core-shell structures via mimicking the Stöber method.

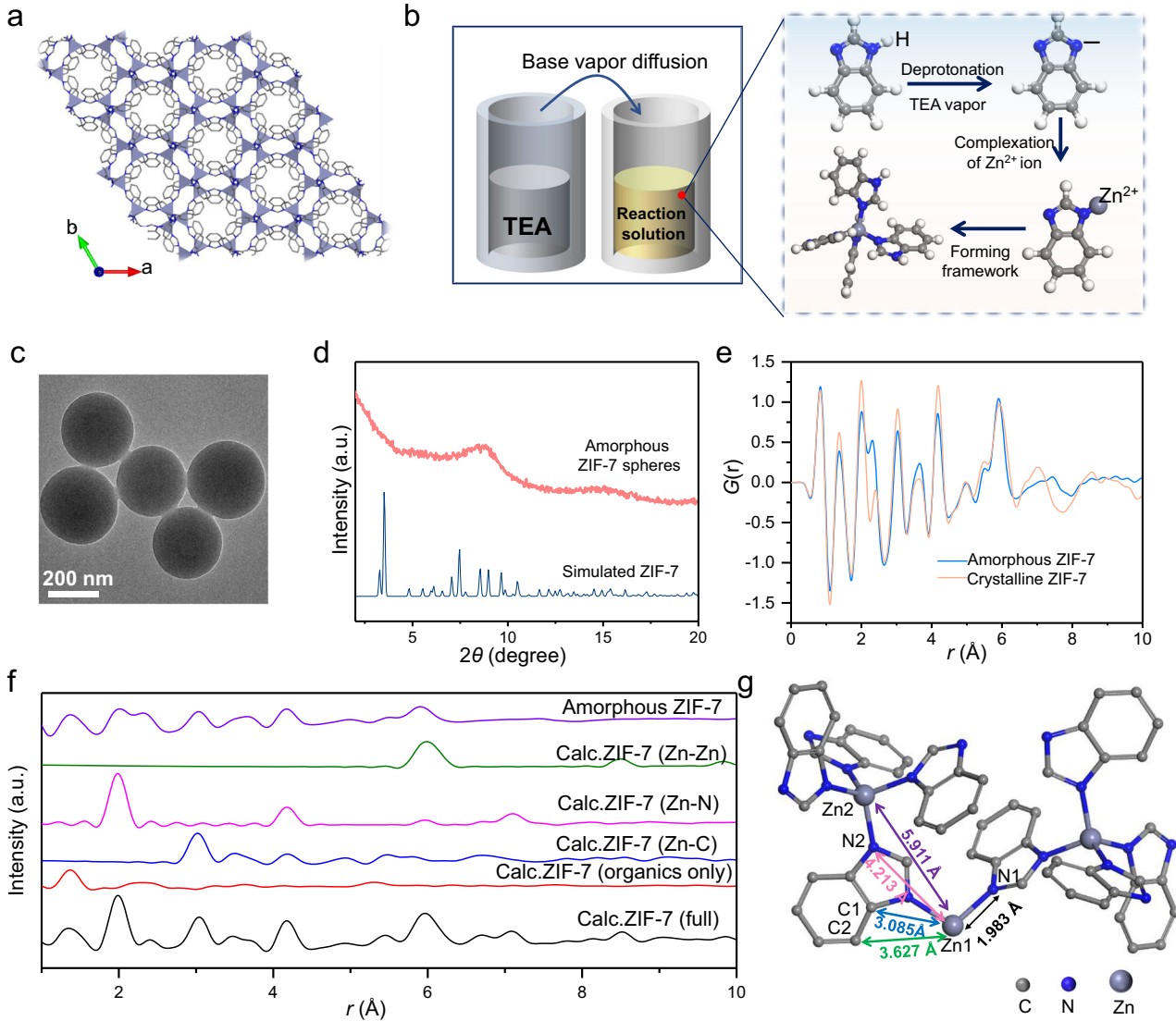

**Fig. 2 | Structural characterizations of aZIF-7 spheres. a** Simplified representation of the ZIF-7 crystal structure. **b** Schematic illustration of the set up and formation process for synthesis of the aZIF-7 spheres. **c** TEM images of aZIF-7 spheres. **d** PXRD pattern of aZIF-7 spheres. **e** Experimental pair distribution functions of aZIF-7 spheres (blue line) and crystalline ZIF-7 (yellow line). **f** Experimental PDF of amorphous ZIF-7 and the calculated patterns of atom-pair distance distribution (Zn-Zn, Zn-C and Zn-N) in ZIF-7. **g** Atomic positions model of ZIF-7. The gray sphere represents carbon atom. The blue sphere represents nitrogen atom. The dark-gray represents zinc atom. Source data are provided as a Source Data file.

adding any TEA, the reaction solution remains transparent for 1 week, even after high-speed and long-term centrifugation, with no precipitation observed. This can be attributed to the stability of the N-H bond in benzimidazole. TEA is a base that promotes the deprotonation of benzimidazole and then the formation of ZIF-7 by complexation of zinc ions with deprotonated benzimidazole molecules. When TEA is added to the MOFs precursors solution, a white precipitate appeared immediately. The Transmission Electron Microscopy (TEM) images (Supplementary Fig. 2) show the formation of agglomerated irregularly shaped nanoparticles. To decrease the deprotonation rate, we varied the concentration of TEA from 0.01 to 0.005, 0.0025, and 0.0005 mol L$^{-1}$. All the obtained samples exhibited a similar agglomeration state (Supplementary Fig. 2). While a reduced concentration of TEA does decrease the nucleation rate, it also hinders the generation of deprotonated benzimidazole as the reaction proceeds. As a result, the nuclei are unable to grow into well-defined colloids. This implies that the synthesis of homogeneous ZIF-7 colloidal particles is not achievable through the direct addition of TEA to the MOFs precursor solution.

Recently, our group has developed a TEA-based gas diffusion method to regulate the nucleation and growth of MOFs.[51] Unlike directly adding TEA, introducing TEA vapor through gas-phase diffusion could reduce the nucleation rate and provide the continuous formation of deprotonated ligand for subsequent growth. Here, we use this approach for synthesizing monodisperse amorphous ZIF-7 spheres. The TEA and the MOFs precursors solutions were firstly placed in two open glass vials, respectively and sealed in a beaker (Fig. 2b). The gradual diffusion of TEA vapor into the mother solution promotes the deprotonation of benzimidazole, enabling the subsequent complexation of Zn$^{2+}$ by benzimidazole to form the ZIF-7 framework. The continuous introduction of triethylamine vapor enables a controlled release of additional deprotonated benzimidazole molecules. Such a controlled release, permits the gradual growth of ZIF-7 nuclei into uniform colloids. The as synthesized ZIF-7 exhibits a monodisperse spherical shape with a size of about 350 nm (Fig. 2c and Supplementary Figs. 3–4). The powder X-ray diffraction (PXRD) pattern exhibits only a broad reflection around 8° indicating the amorphous nature of the ZIF-7 spheres (Fig. 2d). We utilized high-angle

annular dark-field scanning transmission electron microscopy (HAADF-STEM) along with energy-dispersive X-ray spectrometry (EDX) to further study the compositions of the amorphous ZIF-7 (aZIF-7) spheres. Supplementary Fig. 5 shows that Zn, C and N are uniformly distributed throughout the sphere. Compared to the previous diffusion strategy for synthesizing MOFs, this base-vapor diffusion method inherits the typical characteristics of the Stöber method. Firstly, the base vapor in this diffusion strategy is required. The functionality of the base is utilized to control the deprotonation of the organic ligands and to finely tune the kinetics of this sol-gel reaction. Additionally, the growth of the colloid is achieved through continuous crosslinking polymerization. In previous reports, MOFs growth facilitated by vapor diffusion method was achieved through a typical crystallization process, involving nucleation and subsequent crystal growth. This mechanism differs from the continuous crosslinking growth process in this study. Furthermore, colloidal growth is primarily governed by surface energy and is not constrained by the lattice of the material itself. The resulting morphology is spherical in shape, rather than having a Wulff morphology.

To prove that the amorphous spheres possess the same structural characteristics of crystalline ZIF-7, pair distribution function (PDF) is employed to characterize the short-range order in the sample (Fig. 2e). Highly crystalline ZIF-7 was synthesized for comparison. The peaks in the PDF data show nearly identical atom-pair distances for both crystalline ZIF-7 and amorphous spheres, confirming the presence of a short-range order of the ZIF-7 structure within the amorphous spheres. Additionally, the calculated patterns of atom-pair distances of Zn-based bonds are applied to gain insight into the aZIF-7 structure (Fig. 2f, g). The peak of aZIF-7 at a distance of 2 Å corresponds to the nearest Zn-N coordination bond (e.g. Zn1-N1 in Fig. 2f) in the ZIF-7 structure. The distance around 4.2 Å matches the next nearest Zn-N correlation (e.g. Zn1-N2). The link is further confirmed by studying the calculated patterns using Crystallographic Information File (CIF) data, shown in Fig. 2f. There is a general good match between the calculated pattern of ZIF-7 and the experimental aZIF-7. Dissecting the pattern into atom specific correlations between the metal and linker (Zn-C and Zn-N) confirms the coordination between the two with a good agreement between the patterns. This is particularly visible with the nearest neighbor correlations of e.g. Zn1-C1 at 3.08 Å and Zn1-N1 at 1.98 Å. A longer-range order between Zn atoms were also observed at 5.9 Å confirming a smaller network forming in the MOF. To further confirm it, ZIF-7-II, another crystal, was used for comparison because it shares the same composition as ZIF-7 but has a significantly different crystal structure (Supplementary Fig. 6). When compared with ZIF-7-II, the local structure of the crystalline ZIF-7 exhibits a much better fit, with an almost 100% match of peaks up to 6 Å. Especially in the calculated pattern (Supplementary Fig. 7), while some peaks do match between the experimental and the calculated pattern of ZIF-7-II. The overall match is relatively poor. These results demonstrate that aZIF-7 exhibits the same structural features of crystalline ZIF-7 at a short-range.

## Nucleation and growth process

We carried out a sequence of controlled experiments to gain insight into the formation process. First, we change the rate of TEA vapor diffusion by varying the volume ratios of the TEA/ethanol solution (from 1:10, 2:10, 3:9, 6:6, 9:3 to 12:0). At the volume ratios below 9:3, the reaction produced uniform aZIF-7 spheres (Supplementary Fig. 8a–c). Lower TEA concentrations correspond to slower diffusion rates, and thus allow for controlled growth into uniform spherical colloids. When the ratio exceeds 6:6, the corresponding higher TEA diffusion rate results in the occurrence of multiple nucleation events. The reaction product is an agglomerate of non-uniform spheres (Supplementary Fig. 8d–f). These results indicate that slow diffusion/deprotonation is critical for the growth of uniform aZIF-7 spheres. Additionally, we also investigated the effect of reactant concentration.

aZIF-7 spheres size grew from 180 nm to 340 nm when the $ZnCl_2$ concentration was increased from $0.2\,mg\,mL^{-1}$ to $0.6\,mg\,mL^{-1}$ (all constituents in the reaction solution are proportionally scaled) (Supplementary Fig. 9), proving that we can easily tune the size of the spheres by varying the reactant concentration.

We employed time-resolved TEM experiments to study the evolution of the aZIF-7 spheres. After the reactions are proceeded for a given time, various intermediates are sampled for TEM imaging (Supplementary Fig. 10). After 1 h, the reaction solution began to turn turbid and nanoparticles with sizes below 100 nm are formed. As the reaction proceeds, the nanoparticles undergo spherical growth, resulting in a gradual increase in their size from 200 nm to 350 nm. Based on the experimental observations, we propose a potential growth mechanism for aZIF-7 spheres (Supplementary Fig. 11). At the beginning of the reaction, the concentration of deprotonated benzimidazole in the reaction solution was low (pKa = 17.4).[52] As the TEA vapor gradually diffuses into the solution, the concentration of deprotonated benzimidazole increases. aZIF-7 nucleation occurs when the reactant exceeds the supersaturation. During the growth process, as soon zinc-benzimidazole complexes are generated they condensate at the surface of the existing particles to reduce their surface energy. At the end of the growth process monodisperse spherical particles are obtained.

To explain why only amorphous ZIF-7 particles are obtained under our reaction conditions, we dispersed the aZIF-7 spheres in ethanol and then allowed the solution to age for 24 h at 30, 50 and 90 °C, respectively. aZIF-7 incubated at 30 °C remained amorphous, while from 50 °C, the sample starts to crystallize. As the temperature increases, the crystallinity also increases as demonstrated by the sharpening of the reflections in the PXRD pattern (Supplementary Fig. 12). Amorphous ZIF-7 spheres underwent a typical dissolution-recrystallization process during heat treatment (Supplementary Fig. 13). These results indicate that amorphous ZIF-7 is a thermodynamically metastable form (Supplementary Fig. 14), and its formation is controlled by reaction kinetics, this allows to synthesize uniform spherical structures by precisely controlling the reaction rate as in the case of the Stöber method.

## Synthesis of various amorphous MOFs and CPs spheres

To prove the generality of our approach we investigated the synthesis of amorphous MOFs or CPs with different metal nodes and ligands. We chose ZIF-9 (from $Co^{2+}$ and benzimidazole)[53] as a starting candidate because it shares the same crystal structure as ZIF-7, but incorporates different metal nodes (Fig. 3 and Supplementary Fig. 15). Through the TEA vapor diffusion method, a-ZIF-9 spheres were successfully obtained (Fig. 3a and Supplementary Figs. 15–16). Afterwards, we selected systems with significantly different topologies: ZIF-zni $(Zn(imidazole)_2)$[50], SALEM-2 $(Zn(imidazole)(2\text{-methylimidazole}))$,[54] $Zn(2\text{-ethylimidazole})_2$,[50] $Co(2\text{-ethylimidazole})_2$,[55] $Zn(5\text{-chlorobenzimidazole})_2$,[56] $(Co(5,6\text{-dimethylbenzimidazole})_2)$,[57] and $(Co(purine)_2)$[50] each comprised of distinct ligands or even mixed ligands. All the selected organic ligands can be deprotonated by TEA, resulting in the successful formation of the corresponding well-shaped spheres (Fig. 3a, Supplementary Figs. 19–20, 30, 34, 35, 38, 42, 45 and 46). The aforementioned CPs consist of metal ions ($Zn^{2+}$ or $Co^{2+}$) and N-based organic ligands. To further extend the diversity of CPs, we apply this strategy to other metal ions ($Al^{3+}$, $Mn^{2+}$, $Cr^{3+}$, $Fe^{2+}$, $Ni^{2+}$, $In^{3+}$, $Nd^{3+}$, $Eu^{3+}$, $Er^{3+}$ and $Gd^{3+}$) and O or S-based ligands (Fig. 3). TEM images show that the successfully synthesis of diverse amorphous CPs colloids after a similar controlled deprotonation process is achieved (Supplementary Figs. 47, 51, 55, 58–61, 63–70 and 72–73). The XRD patterns of all these spheres display a broad peak, indicating their amorphous nature (Supplementary Figs. 15, 19, 30, 34, 35, 38, 42, 45–47, 51, 55, 58–61, 66–70 and 72–73). These results demonstrate that our proposed method is a versatile synthesis tool for synthesizing chemically diverse amorphous MOFs or CPs spheres.

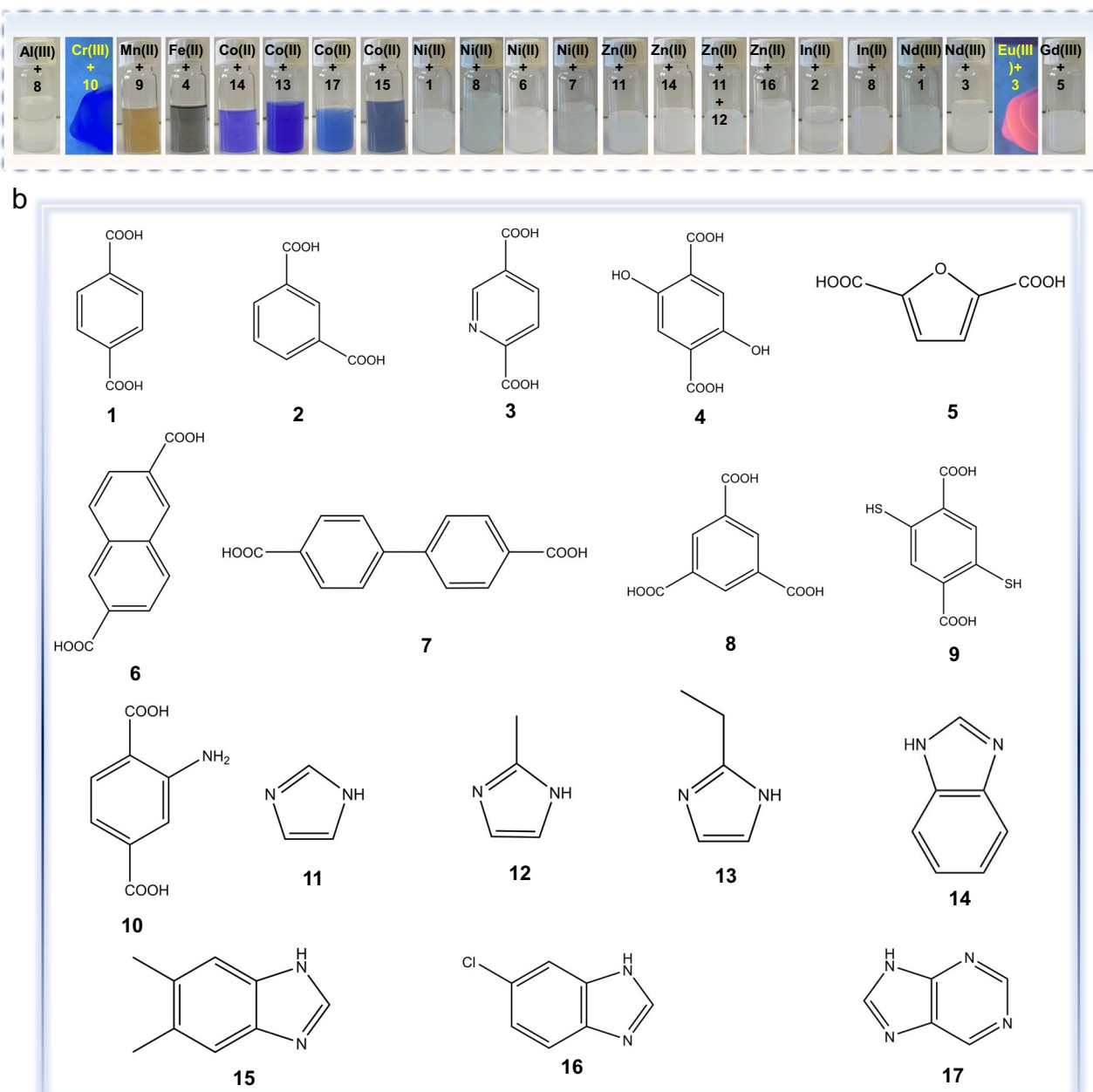

**Fig. 3 | Synthesis of spherical amorphous MOF and CPs colloids. a** The optical images of amorphous MOFs or CPs colloidal solution. **b** 17 types of organic ligands. **1**: Benzene-1,4-dicarboxylic acid, **2**: Benzene-1,3-dicarboxylic acid, **3**: Pyridine-2,4-dicarboxylic acid, **4**: 2,5-Dihydroxyterephthalic acid, **5**: Furan-2,5-dicarboxylic acidic acid, **6**: 2,6-Naphthalenedicarboxylic acid, **7**: Biphenyl-4,4'-dicarboxylate, **8**: Benzene-1,3,5-tricarboxylic acid, **9**: 2,5-disulfhydrylbenzene-1,4-dicarboxylic acid, **10**: 2-Aminoterephthalic acid, **11**: Imidazole, **12**: 2-Methylimidazole, **13**: 2-Ethylimidazole, **14**: Benzimidazole, **15**: 5,6-Dimethylbenzimidazole, **16**: 5-Chlorobenzimidazole, **17**:Purine.

## Growth of conformal amorphous shells on NPs

Amorphous materials are advantageous, compared to their crystalline counterparts, for the conformal coating of nanostructured materials. As a matter of fact, a large number of core-shell colloids with uniform amorphous $SiO_2$ shell have been synthesized based on the Stöber method.[11,58–60] Here, we demonstrate that the TEA vapor diffusion method can also be applied to the synthesis of core-shell structures with an amorphous MOF shell. For example, pre-formed Prussian blue nanocubes cores (PB) can be introduced into the reaction solution for the synthesis aZIF-7 spheres. Under continuous diffusion of TEA, monodisperse PB@aZIF-7 nanocubes are successfully obtained after 1 h reaction time. STEM images and EDX maps show that PB is uniformly confined within the interior part of the core-shell structure

(Fig. 4a, Supplementary Figs. 27d and 29), while, aZIF-7 (Zn-benzimidazole) forms the conformal coating in the outer region. It is worth noting that no separate aZIF-7 spheres were formed under these conditions. This is attributed to the slow rate of TEA diffusion, which preferentially induces heterogeneous nucleation at the surface of the nanoparticles cores instead of homogeneous nucleation. Similar to the formation of aZIF-7 spheres described above, the uniform aZIF-7 coating on PB is obtained by the minimization of the surface energy of the aZIF-7 oligomers.

To further gain insight into the formation process of the core-shell structure, the effect of the pH of the precursors is carefully investigated and two surfactant-free $Fe_2O_3$ particles are selected as core-NPs. The direct addition of TEA into the precursor solution is applied to

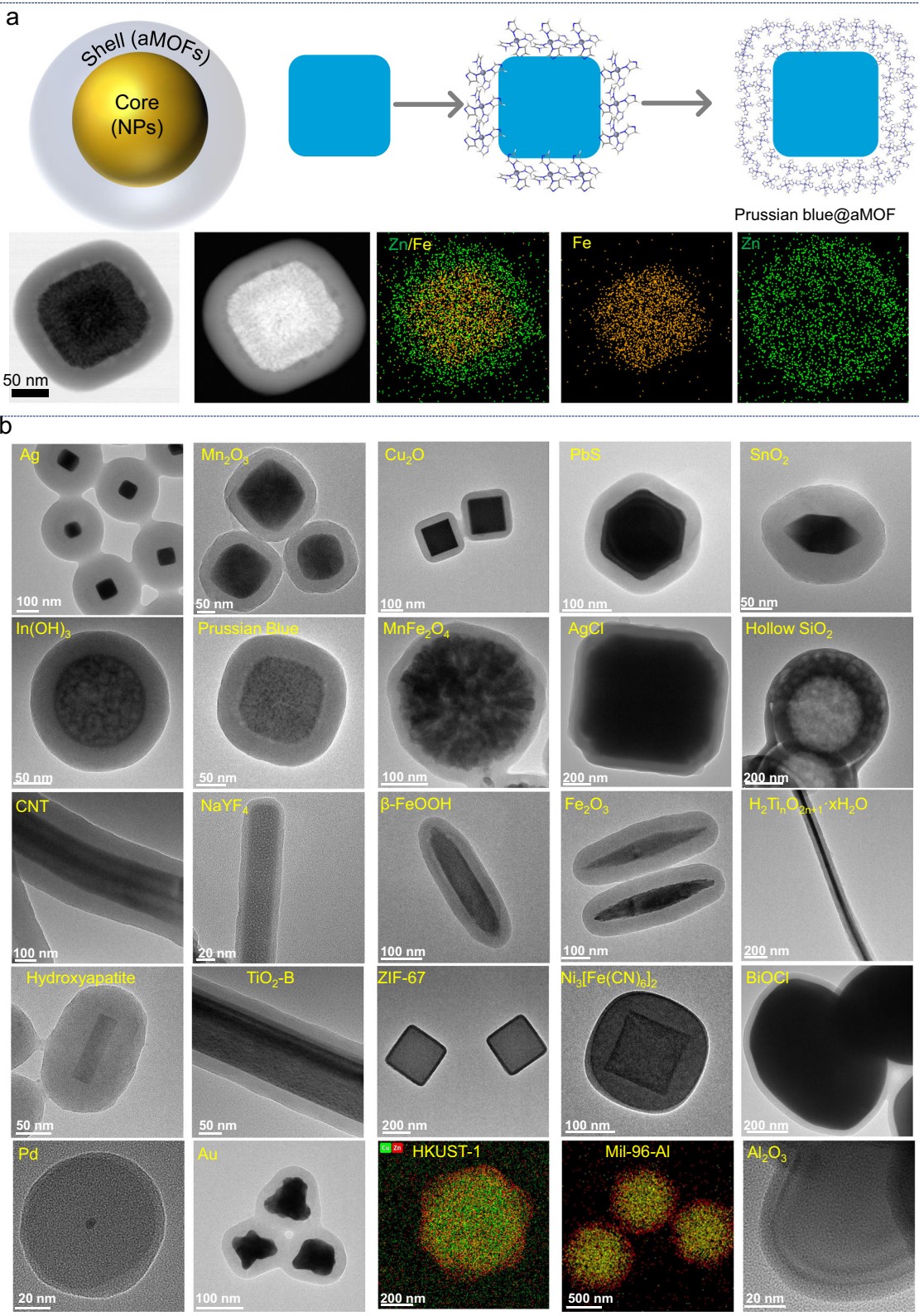

**Fig. 4 | Core-shell colloids (amorphous MOF as shell). a** Schematic illustration of formation process of Prussian Blue@ZIF-7, and STEM and EDX maps. In the element mapping images, yellow represents Zn and green represents Fe atom. **b** TEM images of some selected core-shell particles. Ag@aZIF-zni, Mn$_2$O$_3$@aZn(5-chlorobenzimidazole)$_2$, Cu$_2$O@aZIF-zni, PbS@aZIF-zni, SnO$_2$@aZIF-9, In(OH)$_3$@aZIF-zni, PB@aZIF-7, MnFe$_2$O$_4$@aZIF-7, AgCl@aZIF-zni, Hollow SiO$_2$@aZIF-zni, CNT@aZn(2-ethylimidazole)$_2$, NaYF$_4$@aNi-**8**, β-FeOOH@aZIF-7, Fe$_2$O$_3$@aZIF-zni, H$_2$TiO$_{2n+1}$·xH$_2$O@aZIF-9, HAP@aZn(2-ethylimidazole)$_2$, TiO$_2$@aZIF-7, ZIF-67@aZIF-zni, Ni$_3$[Fe(CN)$_6$]$_2$@aZIF-zni, BiOCl@aZIF-zni, Pd@aZIF-zni, Au@aZIF-zni, HKUST-1@aZIF-zni (red represents Zn and blue represents Cu), MIL-96(Al)@aZIF-zni (red represents Zn and yellow represents Al) and Al$_2$O$_3$@aMOF-74(Fe).

adjust the pH. In Supplementary Figs. 74–75, without any additional TEA added prior to the reaction (pH = 7.45), uniform amorphous ZIF-7 forms around $Fe_2O_3$ particles, and no individual aZIF-7 colloids can be found. This indicates that heterogeneous nucleation dominates during this process. When the pH of the precursor solution reaches 9.35 (with the addition of 2.5 μl TEA), a thin layer of ZIF-7 deposits around the $Fe_2O_3$, but a large number of individual aZIF-7 particles also emerge. A higher pH of the precursor solution accelerates the deprotonation of the ligands, resulting in simultaneous occurrences of homogeneous and heterogeneous nucleation. When the pH increases even further, especially up to 9.90, homogeneous nucleation predominates, with hardly any a-ZIF forming shell structures around the $Fe_2O_3$. These results demonstrate that the slow rate of TEA diffusion in this method play a key role in inducing heterogeneous nucleation (Supplementary Fig. 76).

While certain surfaces, such as zirconia, hematite, and $TiO_2$, demonstrate strong chemical affinity enabling the nucleation of the $SiO_2$ coating, many surfaces require the assistance of surfactants or functional groups for a successful $SiO_2$ coating.[11,58–60] In the present case, the preferred heterogeneous nucleation of CPs at the surface of the core nanoparticles may facilitate the deposition of amorphous CP independently of their surface properties or structures. To verify our hypothesis and the universality of our synthesis method, 20 CPs and over 30 core-nanoparticles including Ag, Au, Pt, Pd, Si, $Fe_2O_3$, Hollow $SiO_2$, $Cu_2O$, $SnO_2$, $Mn_2O_3$, $TiO_2$, $Al_2O_3$, β-FeOOH, $In(OH)_3$, $H_2Ti_nO_{2n+1}$, $MnFe_2O_4$, PbS, AgCl, CdSe, carbon nanotube, $NaYF_4$, BiOCl, $LiNiMnCoO_2$, SiC, Prussian blue, $Ni_3[Fe(CN)_6]_2$, ZIF-67, $Ca_5(PO_4)_3OH$, HKUST-1, MIL-96(Al) and PDA (polydopamine) have been chosen to synthesize a variety of core-shell colloids. The TEM images clearly demonstrate that >100 core-shell colloids have been successfully obtained (Fig. 4b, Supplementary Figs. 17–18, 21–29, 31–34, 36–37, 39–41, 43–44, 48–50, 53–54, 56–59, 62–64, 66–68 and 71). In addition, this amorphous coating technology also hold promise for expansion into thin film synthesis (Supplementary Fig. 77). These results provide ample evidence of the generalizability of our method.

## Shell thickness control

In addition to the generality of our coating method, here we emphasize the tunability of the conformal amorphous coating thickness independently of the morphology of the substrate (Fig. 5a). Ag NPs with different morphologies ranging from cube, tetrahedron to nanowire are introduced into the reaction solution for the synthesis of aZIF-zni. In Fig. 5b, the TEM images clearly illustrate that a uniform and conformal aZIF-zni coating is successfully deposited on Ag NPs independently of their shapes. Furthermore, we can control the thickness of the amorphous MOF shell by simply changing the concentration of the core-NPs. We controlled the initial Ag NPs concentration from 0.1, 0.2, 0.5 to 1 mg $L^{-1}$, while keeping the other conditions unchanged. The corresponding shell thickness is reduced from few tens of nanometers to a few nanometers (Fig. 5c). Even, a uniform encapsulation down to 2.7 nm was successfully achieved on Ag tetrahedrons.

## Applications of the amorphous-based colloids

One potential application of these core-shell colloids lies in their use as sacrificial templates to fabricate multifunctional nanostructures. Yolk-shell structures, while resembling core-shell structures, differentiate themselves by including a void between the core and the shell, which makes it interesting for applications in the fields of catalysis, energy, and drug delivery.[61,62] Here, we propose a general method to synthesize conformal yolk-amorphous MOFs shell structures by using core-amorphous MOFs shell as template. Tannic acid has been demonstrated as an effective agent for selectively etching the inner part of crystalline MOFs to form hollow MOFs.[63] Through the introduction of tannic acid into a solution containing as-prepared core-aMOFs shell colloids, conformal β-FeOOH@void@aZIF-zni with different shape,

PB@void@aZIF-zni, hollow $SiO_2$@void@aZIF-zni, $Mn_2O_3$@void@aZIF-zni and polydopamine@void@aZIF-zni were obtained (Fig. 6a, b and Supplementary Figs. 78–79) and prove the generality of this synthesis route. In addition, core-amorphous MOFs shell colloids can be converted into other functional core-shell particles as well. TEM images show that $Fe_2O_3$@aZIF-zni can be transformed into $Fe_2O_3$@ZnO through calcination in air (Fig. 6c and Supplementary Fig. 80), and Si@Fe(II)-MOF-74 can be successfully converted into Si@C core-shell structures through carbonization (Fig. 6d and Supplementary Fig. 81). These results illustrate the potential of our method to transform core-shell colloids into complex nanostructures for different applications. As a peculiar example, the Si@C core-shell structure with a uniform, conformal, and highly graphitic carbon layer (Supplementary Fig. 81) is highlighted. The carbon coating serves as a protective shell to alleviate the consequences of Si volume change during lithiation and to enhance interface conductivity (Fig. 6d and Supplementary Fig. 82). Due to the large volume change during lithium insertion/extraction,[64] the specific capacity of pure Si decreases to nearly 0 mAh $g^{-1}$ after only 40 cycles (Fig. 6d). In contrast, the Si@C core-shell colloids can still maintain a specific capacity of around 2000 mAh $g^{-1}$ even after 100 cycles.

## Discussion

All in all, our results clearly show that our TEA diffusion method to amorphous MOFs spheres and coatings mimic the Stöber method to silica. The two processes make use of a base to activate the reactions. The Stöber method is based on hydrolysis and condensation reactions and the MOFs synthesis on the controlled deprotonation of the organic ligands and their complexation to metal ions. The rate determining steps are the hydrolysis in the case of the sol-gel process of silica under basic conditions and the deprotonation of the ligands in the TEA diffusion method to amorphous MOFs. The two processes also show some differences that deserve attention. The most notable difference is the crystallization process. For silica, the energy barrier for crystallization is high. Therefore, high temperature is required to convert amorphous silica to a crystalline state. While, compared to the strong Si-O iono-covalent bond, the coordination bond between metal ions and organic ligands are more easily tunable including rotation, break, recombination, enabling the growth process of MOFs to involve transformation from amorphous to crystalline states at lower temperature. It is noteworthy that this method also has its limitations. This strategy is currently not applicable for ligands with low solubility in alcohols at room temperature.

The ordered pore structure and large specific surface area are characteristic features of MOFs. In this work, due to the collapse of the intrinsic ordered framework, the amorphous frameworks exhibit low specific surface area (Supplementary Figs. 83–85 and Supplementary Table 1). However, as colloidal particles transition from the amorphous to crystalline state, the pore volumes increase as well as the gas adsorption properties (Supplementary Fig. 85). The potential applications of these amorphous colloids may arise from following aspects: (1) The characteristics of the corresponding crystalline MOFs is an important criterion to guide the selection. Although lacking porosity and a long-range ordered structure, amorphous colloids still exhibit the typical chemical or physical properties of the crystalline phase. (2) Abundant unsaturated metal sites within the amorphous framework exhibit robust catalytic activity, endowing promising potential for diverse catalytic systems. (3) Amorphous core-shell structures can be carefully converted into crystalline MOFs composites (Supplementary Figs. 86–87). This provides a promising technological pathway for synthesizing crystalline MOF composites.

In summary, we have proposed a mild and versatile strategy for synthesizing well-defined amorphous metal organic framework and coordination polymer spheres and their conformal amorphous coatings by mimicking the classic Stöber method. The slow and continuous diffusion of triethylamine vapor is used to control the deprotonation rate

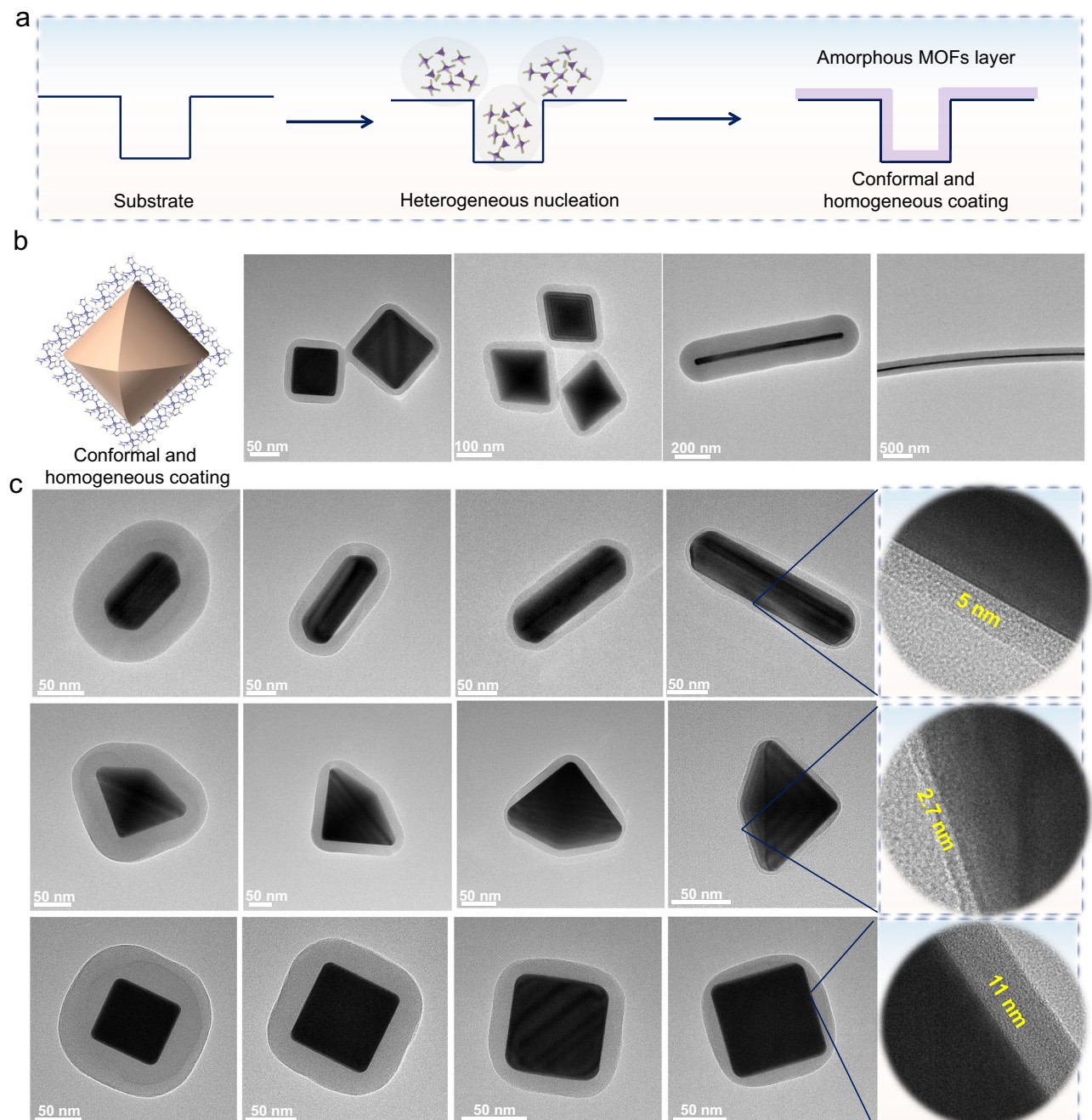

**Fig. 5 | Conformal and homogeneous aMOF coating layer with controllable thickness. a** Schematic illustration of a substrate coated with conformal and homogeneous aMOFs layer. **b** TEM images of Ag@aZIF-zni with different shapes ranging from cube, tetrahedron to nanowire. **c** TEM images of various morphologies of Ag@aZIF-zni with controllable coating thickness.

of the organic ligands, thus allowing for growth of well-defined amorphous CPs spheres. This method is generally applicable and was demonstrated by the synthesis of 24 amorphous CPs spheres made from different metal ions and organic ligands. Furthermore, through the introduction of guest NPs, we have synthesized conformal amorphous CPs coatings leading to uniform core-shell colloids. A gradual deprotonation process facilitates the heterogeneous nucleation of amorphous MOFs on any substrate, regardless of their surface chemistry, structure and morphology. Over 100 core-shell colloids that combines 20 different amorphous MOF or CP shells and >30 different core-NPs have been synthesized with our method. The core-amorphous MOFs shell colloids can be further converted into a wide range of functional colloids, including yolk-amorphous MOFs shell, core@metal oxide, and

core@carbon particles, through simple liquid-phase or solid-state transformation processes. The present work significantly enriches the Stöber method and introduces a platform, enabling the systematic design of colloids exhibiting different level of functionality and complexity.

## Methods

### Synthesis of amorphous ZIF-7 colloids

ZnCl$_2$ (6 mg) and benzimidazole (6 mg) were dissolved in 10 mL of ethanol to form the transparent mother solution in a 40 mL vial left uncapped. Then, a triethylamine solution was prepared by adding 2 mL of TEA to 10 mL of ethanol in a separate 40 mL vial left uncapped. Subsequently, the mother solution and the TEA solution were carefully

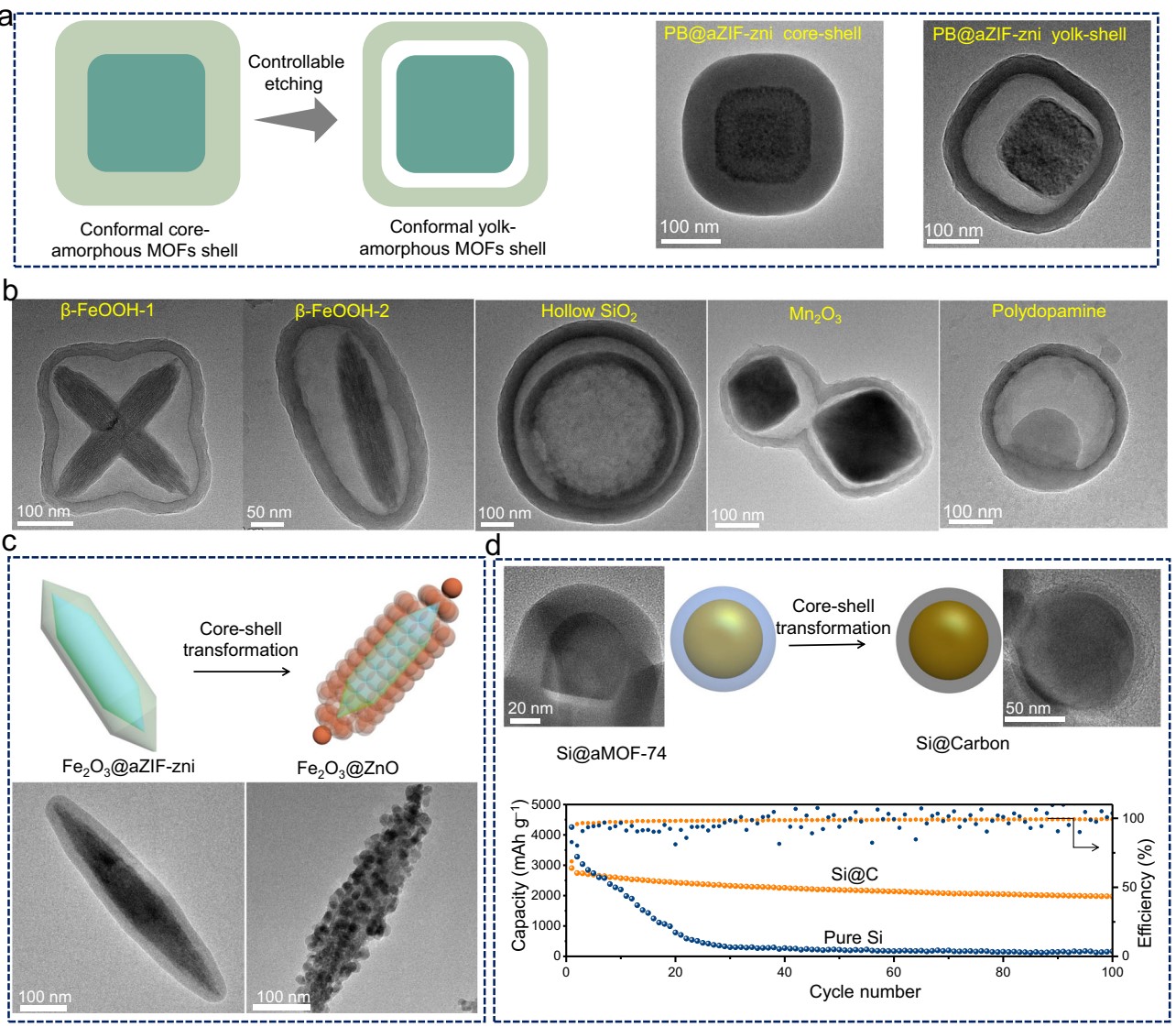

**Fig. 6 | Core@amorphous MOFs shell: sacrificial template to synthesize multifunctional structures. a** Synthesis of conformal yolk-shell structures by etching of core-shell colloids. **b** TEM images of various yolk-shell structures. Conformal β-FeOOH@void@aZIF-zni with different shape, hollow SiO₂@void@aZIF-zni, Mn₂O₃@void@aZIF-zni and polydopamine@void@aZIF-zni. **c** Transformation of Fe₂O₃@aZIF-zni into Fe₂O₃@ZnO. **d** Transformation of Si@aMOF-74-Fe into Si@Carbon core-shell colloids and the cycling performance of the obtained Si@Carbon electrode in lithium ion battery. Source data are provided as a Source Data file.

sealed at room temperature in a 500 mL beaker. Throughout the reaction, the mother solution was gently stirred. The TEA vapor generated from the TEA solution gradually diffused into the mother solution. The amorphous ZIF-7 spheres were collected after 4 h and washed with absolute ethanol for three times.

## Synthesis of core-shell structures (amorphous ZIF-7 as shell)

The synthesis protocol of core-shell structures is similar to that of amorphous ZIF-7 spheres. Briefly, ZnCl₂ (6 mg) and benzimidazole (6 mg) were dissolved in 10 mL of ethanol to form the transparent mother solution in a 40 mL vial left uncapped followed by the addition of 1 mg of core-NPs seeds. Then, a TEA solution was prepared by adding 2 mL of TEA to 10 mL of ethanol in a separate 40 mL vial left uncapped. Subsequently, the mother solution and the TEA solution were carefully sealed at room temperature in a 500 mL beaker. Throughout the reaction, the mother solution was gently stirred. The TEA vapor generated from the TEA solution gradually diffused into the mother solution. The core-shell colloids were obtained after 1 h and washed with absolute ethanol for three times.

## Synthesis of amorphous ZIF-9 colloids

CoCl₂·6H₂O (13.5 mg) and benzimidazole (9 mg) were dissolved in 10 mL of ethanol to form transparent mother solution in a 40 mL vial. It is closed using a cap with three holes (1 mm diameter). Then, a TEA solution was prepared by adding 3 mL of TEA to 9 mL of ethanol in a separate 40 ml vial. A cap punctured by three holes was used to close the TEA solution. Subsequently, the mother solution and the TEA solution were carefully sealed at room temperature in a 500 mL beaker. Throughout the reaction, the mother solution was gently stirred. The TEA vapor generated from the TEA solution gradually diffused into the mother solution. The amorphous ZIF-9 spheres were collected after 5 h and washed with absolute ethanol for three times.

## Synthesis of core-shell structures (amorphous ZIF-9 as shell)

The synthesis protocol of core-shell structures is similar to that of amorphous ZIF-9 spheres. Typically, CoCl₂·6H₂O (13.5 mg) and benzimidazole (9 mg) were dissolved in 10 mL of ethanol to form transparent mother solution in a 40 mL vial followed by the addition of the core-NPs (0.5 mg). It is closed using a cap with three holes (1 mm

diameter). Then, a TEA solution was prepared by adding 3 mL of TEA to 9 mL of ethanol in a separate 40 mL vial. A cap punctured by three holes was used to close TEA solution. Subsequently, the mother solution containing the core-nanoparticles and TEA solution were carefully sealed at room temperature in a 500 mL beaker. Throughout the reaction, the mother solution was gently stirred. The amorphous ZIF-9-based core-shell colloids were obtained after 3 h and washed with absolute ethanol for three times.

## Synthesis of amorphous MOF-74(Fe) colloids

$FeCl_2 \cdot 4H_2O$ (6 mg) and 2,5-Dihydroxyterephthalic acid ($H_4$dobdc, 2.6 mg) were dissolved in 10 mL of methanol to form transparent mother solution in a 40 mL vial left uncapped. Then, a TEA solution was prepared by adding 1 mL of TEA to 11 mL of ethanol in a separate 40 mL vial left uncapped. Subsequently, the mother solution and the TEA solution were carefully sealed at room temperature in a 500 mL beaker. Throughout the reaction, the mother solution was gently stirred. The TEA vapor generated from the TEA solution gradually diffused into the mother solution. The amorphous MOF-74(Fe) spheres were collected after 2 h and washed with absolute ethanol for three times.

## Synthesis of core-shell structures (amorphous MOF-74(Fe) as shell)

The synthesis protocol of core-shell structures is similar to that of amorphous MOF-74(Fe) spheres. $FeCl_2 \cdot 4H_2O$ (6 mg) and 2,5-Dihydroxyterephthalic acid ($H_4$dobdc, 2.6 mg) were dissolved in 10 mL of methanol to form transparent mother solution in a 40 mL vial left uncapped followed by the addition of the core-NPs (2 mg). Then, a TEA solution was prepared by adding 1 mL of TEA to 11 mL of ethanol in a separate 40 mL vial left uncapped. Subsequently, the mother solution and the TEA solution were carefully sealed at room temperature in a 500 mL beaker. Throughout the reaction, the mother solution was gently stirred. The TEA vapor generated from the TEA solution gradually diffused into the mother solution. The amorphous MOF-74(Fe)-based core-shell colloids were collected after 2 h and washed with absolute ethanol for three times.

## Synthesis of amorphous Ni-6 colloids

$NiCl_2 \cdot 6H_2O$ (4 mg) and 2,6-Naphthalenedicarboxylic acid ($H_2$ndc, 4 mg) were dissolved in a mixture of 7 mL of ethanol and 3 ml of DMF to form transparent mother solution in a 40 mL vial left uncapped. Then, a TEA solution was prepared by adding 1 mL of TEA to 11 mL of ethanol in a separate 40 mL vial left uncapped. Subsequently, the mother solution and TEA solution were carefully sealed at room temperature in a 500 mL beaker. Throughout the reaction, the mother solution was gently stirred. The TEA vapor generated from the TEA solution gradually diffused into the mother solution. The amorphous Ni-6 spheres were collected after 3 h and washed with absolute ethanol for three times.

## Synthesis of core-shell structures (amorphous Ni-6 as shell)

The synthesis protocol of core-shell structures is similar to that of amorphous Ni-6 spheres. $NiCl_2 \cdot 6H_2O$ (4 mg) and 2,6-Naphthalenedicarboxylic acid ($H_2$ndc, 4 mg) were dissolved in a mixture of 7 mL of ethanol and 3 ml of DMF to form transparent mother solution in a 40 mL vial left uncapped followed by the addition of the core-NPs (2 mg). Then, a TEA solution was prepared by adding 1 mL of TEA to 11 mL of ethanol in a separate 40 mL vial left uncapped. Subsequently, the mother solution and TEA solution were carefully sealed at room temperature in a 500 mL beaker. Throughout the reaction, the mother solution was gently stirred. The TEA vapor generated from the TEA solution gradually diffused into the mother solution. The amorphous Ni-6 based core-shell colloids were collected after 3 h and washed with absolute ethanol for three times.

## Preparation of yolk-amorphous MOFs shell colloids

In a typical synthesis, 10 mg of the as-obtained core@aZIF-zni colloids were added to 2 ml of $H_2O$, followed by sonication for 5 min. Then, 0.3 mL of tannic acid aqueous solution (40 g $L^{-1}$) was added into the above solution and sonicated for 5 min. The yolk-amorphous MOFs shell structures were collected by centrifugation.

## Data availability

The work's findings are backed by data presented in the article and its Supplementary Information files. The data generated in this study are provided in the Source Data file and can be obtained from the corresponding authors upon request as well. Source data are provided with this paper.

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

## Acknowledgements

W.Z acknowledge the Alexander von Humboldt Foundation, Bonn, Germany, for a postdoctoral fellowship and research grant and the Max Planck Society. We thank the help of Carsten Prinz for the gas-sorption test. Prof. Dr. Markus Antonietti is acknowledged for critical reading of the manuscript and fruitful discussions. We acknowledge DESY (Hamburg, Germany), a member of the Helmholtz Association HGF, for the provision of experimental facilities. Parts of this research were carried out at PETRA III beamline P02.1. Beamtime was allocated by an In-House contingent. We acknowledge support by the Open Access Publication Fund of Humboldt-Universität zu Berlin.

## Author contributions

W.Z. and N.P. conceived the project; The synthesis, microscopy, PXRD measurement and the related data analysis was performed by W.Z.; Battery test was performed by Y.C.L; H.S.J performed PDF measurements and analysis. W.Z. and N.P. wrote the manuscript. All authors have approved the manuscript.

## Funding

## Competing interests

The authors declare no competing interests.
