## [Peer Review File · Nature Communications]

Stöber Method to Amorphous Metal-Organic Frameworks and Coordination PolymersREVIEWER COMMENTS

Reviewer #1 (Remarks to the Author):

Inspired by the Stöber method, a protocol has been proposed to generate amorphous metal-organic framework (MOF) spheres and use them to synthesize a variety of hetero-hierarchical structures. Although the method is simple - just controlling the diffusion of triethylenediamine - it is an interesting synthetic method as it allows the synthesis of a wide range of beautiful structures. The products have been carefully identified by electron microscopy, elemental mapping and X-ray, and the experimental results are reliable. Questions and suggestions are given below.

Regarding early work, there is a large number of published work on the synthesis of crystalline/amorphous MOFs of nanospheres, as exemplified by Oh and Mirkin (*Angew Chem Int Ed Engl* 45(33): 5492-5494). The same applies to hierarchical MOF composites (e.g. P. Falcaro, *Acc Chem Res* 2014 Vol. 47 Issue 2 Pages 396-405). The authors should provide a more appropriate and comprehensive introduction to previous reports in the introduction.

Figure 2 shows the PDF analysis, but it is difficult to consider the amorphous as similar to ZIF-7 with this result. MOFs with the same composition but different crystal structures should be taken from the database and compared. If possible, the same PDF analysis should be performed to show that it is different from one of the examples in Figure 2E. Other examples are the same. The composition of the amorphous phase should be identified by several analyses (e.g. elemental analysis, gas adsorption) to confirm that the name of the corresponding MOF crystal can be used to name each amorphous state.

The hierarchical structures composed of different crystals are also of interest. Is it possible to synthesize an amorphous composite using the present method and then promote crystallization to create hierarchical structures of heterocrystals? The result that amorphous ZIF-7 crystallises after treatment at 50°C is interesting and can this behaviour be successfully exploited for the crystal-crystal heterocomposites? Related experiments are necessary.

As a function, only the electrochemical evaluation after conversion to carbon shown in Figure 6D is unsatisfactory. For example, the architecture in Figure 5 is interesting, but we would like to see the unique adsorption properties as a function. For example, the photothermal properties of the internal Ag and the release of guests from the external surface MOF (e.g. *J Am Chem Soc* 135(30): 10998-11005) is one of the ideas to be tested.

Figure 1C and Figure 3A are not needed in the text.

Reviewer #2 (Remarks to the Author):

The manuscript submitted by Zhang and co-workers introduces a novel vapor diffusion method for synthesizing amorphous metal-organic frameworks (MOFs) with controllable compositions and sizes. The chelation kinetics of metal ions and deprotonated ligands are modulated by triethylamine (TEA) vapor, resulting in the controllable growth of MOF particles and coatings, which has been challenging. Specifically, this method successfully synthesized twenty-two different amorphous MOF colloids and over 100 core-shell particles with amorphous MOF shells. Additionally, the authors demonstrated that the core-shell particles could be further transformed into various nanostructures, such as core@metal oxides, through suitable post-modification, underscoring the significant potential of this strategy for generating a new material library. Overall, this manuscript is engaging, well-structured, and provides a new pathway for producing amorphous MOFs, but it requires further polishing as some important information is missing. Therefore, I recommend publication after addressing the following concerns:

Major Comments:

1. I concur that the "Stöber Method" has been widely developed for synthesizing SiO₂, resorcinol-

formaldehyde, and TiO₂, in which precursor solutions usually undergo hydrolysis and condensation (covalent crosslinking). Controlling the coordination kinetics of MOFs is important and has been well-studied but often not considered as the Stöber method. Specifically, the vapor diffusion method has been previously applied to synthesizing MOFs (e.g., *J. Mater. Chem. A*, 2016, 4, 10345-10351), especially cyclodextrin (CD)-MOFs. What type of vapor diffusion method is considered the "Stöber Method"? More clarification on why the current method is labeled as the "Stöber Method" is needed if the authors prefer to keep this term.

2. Pore sizes and surface area are pivotal parameters for MOF materials. Given the nearly identical atom-pair distance and structural features of amorphous MOFs compared to crystalline MOFs (Figure 2), how are the pore sizes and surface areas of amorphous MOFs in this work? Additionally, it would be beneficial to explore whether their sizes and surface areas change with increased crystallinity under heat treatments (Figure S9).

3. Pre-surface modifications are typically required for growing MOFs on particles due to their weak interactions (*ACS Nano* 2015, 9, 7, 6951-6960; *Nat. Chem.*, 2012, 4, 310-316). This work demonstrated an interesting, versatile growing of amorphous MOFs on diverse particles and claimed that the slow rate of TEA diffusion induces the MOF precursors to form heterogeneous nucleation on the surface of nanoparticles instead of homogeneous nucleation. Does the pH of the precursor solution govern the heterogeneous or homogeneous nucleation? More deep investigations are highly recommended, and understanding the dominant interactions between amorphous MOFs and representative core particles would also enhance the fundamental insights provided.

4. Is this amorphous coating technology also applied to planar substrates?

Minor Comments:

1. Abstract: The authors claimed that the current challenge is to "achieving controllable synthesis," which is too broad and unclear. It is better to refine this central sentence to highlight the impact and novelty of the current work more clearly.

2. Introduction, Page 2: The authors mentioned that "Recently, the Stöber method has been successfully extended to other materials including TiO₂ and resorcinol-formaldehyde (RF), 13...", in which both examples were published over 10 years ago. It would be great to highlight more recent advancements in new material composition design based on the Stöber method.

3. Figure 6b: Amorphous MOF coatings retain the similar shapes of the various substrates in Figures 5 and 6 except the b-FeOOH-1-based template (Figure 6B). Please provide a detailed explanation for clarity.

4. Many scale bars in microscopy images (e.g., Figure S2) are not easily visualized. Increase font size and bar thickness for better visibility, if possible.

Reviewer #3 (Remarks to the Author):

This manuscript introduced a general synthesis route to amorphous MOFs using a vapor diffusion method, which allows for the precise control of growth kinetics. The authors successfully synthesized 22 different amorphous MOF colloids by selecting 11 metal ions and 17 organic ligands. Additionally, they demonstrated the versatility of their amorphous coating technology by synthesizing over 100 new core-shell composites from 19 different amorphous MOF shells and 30 different core-NPs. This work significantly enriches the Stöber method and provides a platform for systematically designing colloids exhibiting varying levels of functionality and complexity. Overall, this is an interesting and valuable contribution to the field of MOF synthesis and their potential applications in energy storage, conversion, and drug delivery. Therefore, I recommend acceptance of the manuscript after the following issues have been addressed.

1 Is it possible to replace the triethylamine with commonly used ammonia solution in the Stöber

method as the base? Perhaps this approach could reduce costs.

2 The pore structure is one of the key characteristics of MOF materials. Does this amorphous MOF material still retain its pore structure? The author can characterize one of the samples to help readers better understand the properties of the material.

3 Figure 1c, some shape models do not conform to organic ligands, please confirm it.

4 Page 5 "As the temperature increases, the crystallinity also increases as demonstrated by the sharpening of the reflections in the PXRD pattern (Figure S9)". The author has studied that increasing temperature can improve crystallinity, but has the morphology changed during this process? It is recommended to characterize whether the sample with increased crystallinity still maintains its original morphology.

5 Some writing errors need to be checked, such as line 3, page 13, "activate".

Reviewer #4 (Remarks to the Author):

In this communication, Zhang, Pinna et al. report on a simple strategy that mimics the Stöber method to synthesize diverse amorphous MOFs (aMOFs) spheres and aMOFs based core shell particles. The strategy is based on a TEA-based gas diffusion method to grow amorphous MOF colloids, including core-shell particles. The authors successfully synthesized a large number of new colloids using this original approach.

Personally, I must express my appreciation for this study and the innovative approach proposed by the authors. The idea is original and offers a general method to synthesize various amorphous colloids. The study is well-organized, and the conclusions are well-supported by the data, primarily from microscopy but also incorporating some PDF measurements. I highly recommend its publication in Nature Communications.

However, I do have a few questions regarding the potential applications of amorphous MOFs and how the authors plan to utilize this method in various applications:

Given that the materials are amorphous and lack porosity and order, what criteria guide the selection of a specific MOF?

In this case, can we define the shell as a MOF, or is it more accurately described as a coordination polymer?

Is it possible to achieve similar results with any ligand or metal?

Are there any differences in the mechanical stability of the shell depending on the ligand or metal used?

Additionally, some long-range SEM images would be beneficial to support the homogeneity of each sample.

These questions are raised in the spirit of enhancing the clarity and completeness of the study, and I look forward to the authors' insights on these aspects.

RESPONSE TO REVIEWERS' COMMENTS

General response and changes

Thank you very much for constructive comments on our manuscript. These comments are very helpful for improvement of our manuscript. We basically revised our manuscript on the basis of reviewers' suggestion. The revised parts are highlighted.

Reviewer 1:

General comments: Inspired by the Stöber method, a protocol has been proposed to generate amorphous metal-organic framework (MOF) spheres and use them to synthesize a variety of hetero-hierarchical structures. Although the method is simple-just controlling the diffusion of triethylenediamine - it is an interesting synthetic method as it allows the synthesis of a wide range of beautiful structures. The products have been carefully identified by electron microscopy, elemental mapping and X-ray, and the experimental results are reliable. Questions and suggestions are given below.

Reply: Thanks for your comments. According to your suggestions, we added more data and discussions to help enhance the robustness of our manuscript. Please find the point to point response below.

Comment 1: Regarding early work, there is a large number of published work on the synthesis of crystalline/amorphous MOFs of nanospheres, as exemplified by Oh and Mirkin (Angew Chem Int Ed Engl 45(33): 5492-5494). The same applies to hierarchical MOF composites (e.g. P. Falcaro, Acc Chem Res 2014 Vol. 47 Issue 2 Pages 396-405). The authors should provide a more appropriate and comprehensive introduction to previous reports in the introduction.

Reply: According to your suggestions, we integrated additional discussion about the synthesis of crystalline/amorphous MOFs spheres and crystalline-hierarchical MOF composites in the introduction part. The related contents and references are added and listed as follows:

“In recent years, many amorphous CPs ranging from colloidal to bulk-glass state have been designed and attracted increasing attention due to some distinctive characteristics compared to crystalline CPs. Firstly, the application of crystalline CPs into functional macroscopic systems, such as thin films coatings and bulk materials, is considerably challenged by their inherent brittleness and rigidity. Compared to crystalline CPs, the flexible and disordered nature of amorphous CPs

could effectively alleviate these drawbacks and increase the mechanical and chemical robustness. In addition, amorphous MOFs and CPs have proven to be advantageous in terms of loading of guest molecules and in finely tuning the release characteristics when used in bio-medicine and catalysis.

The synthesis of nano-/microparticles of amorphous CPs have been regarded as the effective strategy to enhance their performance. While some pioneering studies have reported the synthesis of amorphous CPs sphere, the synthetic methods are often tailored to specific systems, and more universally applicable strategy are still required. Furthermore, compared to crystalline MOFs, less attention has been given to the synthesis of hierarchical amorphous MOFs composites. Therefore, the development of an effective a general approach for the synthesis of amorphous CPs and their composites is an important addition to the field of CPs and is also expected to demonstrate novel applications in the fields of catalysis, drug release, and electronics.”

3. Oh, M. & Mirkin, C. A. Chemically tailorable colloidal particles from infinite coordination polymers. *Nature* **438**, 651-654 (2005).
46. Doherty, C. M. *et al.* Using functional nano-and microparticles for the preparation of metal–organic framework composites with novel properties. *Accounts Chem. Res.* **47**, 396-405 (2014).
47. Zhou, J. *et al.* Versatile core–shell nanoparticle@ metal–organic framework nanohybrids: Exploiting mussel-inspired polydopamine for tailored structural integration. *ACS Nano* **9**, 6951-6960 (2015).
48. Oh, M. & Mirkin, C. A. Ion exchange as a way of controlling the chemical compositions of nano-and microparticles made from infinite coordination polymers. *Angew. Chem. Int. Ed.* **45**, 5492-5494 (2006).

Comment 2: Figure 2 shows the PDF analysis, but it is difficult to consider the amorphous as similar to ZIF-7 with this result. MOFs with the same composition but different crystal structures should be taken from the database and compared. If possible, the same PDF analysis should be performed to show that it is different from one of the examples in Figure 2E. Other examples are the same. The composition of the amorphous phase should be identified by several analyses (e.g. elemental analysis, gas adsorption) to confirm that the name of the corresponding MOF crystal can be used to name each amorphous state.

Reply: According to your suggestions, ZIF-7-II, another crystal, was used for comparison because it shares the same composition as ZIF-7 but has a significantly different crystal structure (**Figure S6a**). As shown in **Figure S6b**, when compared with ZIF-7-II, the local structure of the crystalline ZIF-7 exhibits a much better fit, with an almost 100% match of peaks up to 6Å. Especially in the calculated pattern (**Figure S7**), while some peaks do match between the experimental and the calculated pattern of ZIF-7-II. The overall match is relatively poor. In addition, ZIF-7 exhibit a unique gate open behavior in CO₂ adsorption-desorption. When the amorphous ZIF-7 is treated at 90°C, it also exhibits typical gate-opening behavior (**Figure S85**). These results could confirm that the amorphous colloids share identical local structure with ZIF-7. Due to the limited availability of beamline and high number of amorphous colloids in this work, we need more time in the future to study the local structure properties of additional amorphous colloids. To avoid unnecessary misunderstandings, many amorphous MOFs this case will be directly named using their components in such as Al-BTC, Ni-NDC, Ni-BDC, Mn-DSBDC.

The related figures are added as follows:

Figure S6. (a) Simplified representation of the ZIF-7 and ZIF-II crystal structure. (b) Experimental PDF of amorphous ZIF-7, crystalline ZIF-7 and crystalline ZIF-7-II. The ZIF-7 and ZIF-7-II share some chemical compositions, but quite different crystal structure. Compared with ZIF-7-II, a much better fit of the local structure is observed for the ZIF-7 structure. There is an almost 100% match of peaks up to 6Å.

Figure S7. Experimental PDF of amorphous ZIF-7 and calculated patterns of atom-pair distance distribution of ZIF-7-II. While some peaks do match between the experimental and the calculated pattern of ZIF-7-II. The overall match is relatively poor.

Figure S85. The CO_2 adsorption-desorption isotherm collected at 298 K of a-ZIF-7 and its corresponding crystalline product under heat treatment (90°C). The crystalline ZIF-7 exhibits a typical gate-open behavior in CO_2 adsorption-desorption

Comment 3: The hierarchical structures composed of different crystals are also of interest. Is it possible to synthesize an amorphous composite using the present method and then promote crystallization to create hierarchical structures of heterocrystals? The result that amorphous ZIF-7 crystallizes after treatment at 50°C is interesting and can this behaviour be successfully exploited for the crystal-crystal hetero-composites? Related experiments are necessary.

Reply: Based on your suggestions, we have investigated the conversion process of β -FeOOH@amorphous ZIF-7 under treatment at 50 °C. The amorphous-based composite can be successfully converted into crystalline ZIF-7/ β -FeOOH composites (**Figure S86**) under thermal treatment, which was also indicated by PXRD patterns (**Figure S87**). I agree with the reviewer's perspective, this phenomenon of amorphous transformation is interesting. Our experimental results preliminarily confirm that this conversion provides a new pathway for synthesizing crystalline MOF composites. Next step, we will conduct more detailed investigations to achieve precise control over the transition from amorphous to crystalline, enabling accurate modulation of the transformation process. The ultimate objective is to achieve the transition from amorphous to crystalline while preserving the initial morphology.

β -FeOOH@crystalline-ZIF-7

Figure S86. The TEM images of corresponding β -FeOOH/crystalline ZIF-7 composites under heat treatment (50 °C).

Figure S87. The PXRD patterns of corresponding β -FeOOH/crystalline ZIF-7 composites under heat treatment (50 °C).

Comment 4: As a function, only the electrochemical evaluation after conversion to carbon shown in Figure 6D is unsatisfactory. For example, the architecture in Figure 5 is interesting, but we would like to see the unique adsorption properties as a function. For example, the photothermal properties of the internal Ag and the release of guests from the external surface MOF (e.g. J Am Chem Soc 135(30): 10998-11005) is one of the ideas to be tested.

Reply: In this work, a universal synthetic method had been proposed and numerous novel functional architectures had been synthesized. In the next step, we will focus on integrating these functional colloids into specific applications. As pointed out by the reviewer, the core-shell composite depicted in Figure 5 is an intriguing material with promising potential in terms of gas adsorption properties. We will carefully investigate the transition from amorphous composites to crystal-crystal hetero-composites, especially in their gas adsorption-related properties. Although our lab is currently unable to conduct this application (e.g. J Am Chem Soc 135(30): 10998-11005), we plan to seek collaboration for further exploration in the future. We believe that comparative studies between crystalline and amorphous composites can better reveal the advantages in this photothermal gas-adsorption application.

Comment 5: Figure 1C and Figure 3A are not needed in the text.

Reply: According to your suggestions, we have revised these two figures as follows.

Figure 1. Schematic illustration of the synthesis of aMOFs and aCPs colloids. (A) The Stöber method for the synthesis of monodisperse SiO_2 spheres (TEOS=Tetraethyl orthosilicate). (B) The synthesis of aMOFs colloids and core-shell structures via mimicking the Stöber method.

Figure 3. Synthesis of spherical amorphous MOF and CPs colloids. (A) The optical images of amorphous MOFs or CPs colloidal solution. (B) 17 types of organic ligands.

Reviewer 2

General comments: The manuscript submitted by Zhang and co-workers introduces a novel vapor diffusion method for synthesizing amorphous metal–organic frameworks (MOFs) with controllable compositions and sizes. The chelation kinetics of metal ions and deprotonated ligands are modulated by triethylamine (TEA) vapor, resulting in the controllable growth of MOF particles and coatings, which has been challenging. Specifically, this method successfully synthesized twenty-two different amorphous MOF colloids and over 100 core-shell particles with amorphous MOF shells. Additionally, the authors demonstrated that the core-shell particles could be further transformed into various nanostructures, such as core@metal oxides, through suitable post-modification, underscoring the significant potential of this strategy for generating a new material library. Overall, this manuscript is engaging, well-structured, and provides a new pathway for producing amorphous MOFs, but it requires further polishing as some important information is missing. Therefore, I recommend publication after addressing the following concerns:

Reply: Thank you for appreciating the quality of our work. According to your suggestions, we added more data and discussions to help enhance the robustness of our approach. We wish our revised manuscript has now reached the required quality for publication.

Major Comments:

Comment 1: I concur that the "Stöber Method" has been widely developed for synthesizing SiO₂, resorcinol-formaldehyde, and TiO₂, in which precursor solutions usually undergo hydrolysis and condensation (covalent crosslinking). Controlling the coordination kinetics of MOFs is important and has been well-studied but often not considered as the Stöber method. Specifically, the vapor diffusion method has been previously applied to synthesizing MOFs (e.g., J. Mater. Chem. A, 2016, 4, 10345-10351), especially cyclodextrin (CD)-MOFs. What type of vapor diffusion method is considered the "Stöber Method"? More clarification on why the current method is labeled as the "Stöber Method" is needed if the authors prefer to keep this term.

Reply: I agree with the reviewer's suggestion to further discuss and clarify the similarities between the synthetic method in this work and the Stöber method. As you mentioned above, the Stöber method involves hydrolysis step followed by polymerization. In the classic Stöber method, there are some typical features: 1) the rate of hydrolysis is controlled by using a base in an alcohol-based system, thus achieving the regulation of the colloidal synthesis. 2) Unlike the classical

crystallization process, the growth of colloids involves the continuous crosslinking polymerization of precursors. Finally, the sol-gel process of the Stöber method is predominated by surface energy, thus the resulting colloids are all spherical in shape, by minimizing the surface energy.

For mimicking the Stöber method to synthesize aMOFs, a base vapor in this diffusion strategy is required. Therefore, we aim to enhance clarity by using a base-vapor diffusion method. In the case of cyclodextrin (CD)-MOFs, MeOH is vapor, and its function is not to assist in deprotonation process, but to stabilize the crystal structure and promote crystallization. Additionally, colloids growth is form by the continuous crosslinking polymerization in this work. In other cases (e.g., *J. Mater. Chem. A*, 2016, 4, 10345-10351), the MOFs growth facilitated by vapor diffusion method is a classical crystallization process, involving nucleation and subsequent crystal growth. This mechanism differs from the continuous crosslinking growth process of the Stöber method and the amorphous MOFs synthesized in this study. Furthermore, colloidal growth is primarily governed by surface energy and is not constrained by the lattice of the material itself. The resulting morphology is spherical in shape, rather than having a Wulff morphology. The related discussion and references are added as follows:

“Compared to the previous diffusion strategy for synthesizing MOFs, this base-vapor diffusion method inherits the typical characteristics of the Stöber method. Firstly, the base vapor in this diffusion strategy is required. The functionality of the base is utilized to control the deprotonation of the organic ligands and to finely tune the kinetics of the sol-gel reaction. Additionally, the growth of the colloids is achieved by a continuous crosslinking polymerization. In previous reports, MOFs growth facilitated by vapor diffusion method was achieved by a typical crystallization process, involving nucleation and subsequent crystal growth. This mechanism differs from the continuous crosslinking growth process in this study. Furthermore, colloidal growth is primarily governed by surface energy and is not constrained by the lattice of the material itself. The resulting morphology is spherical in shape, rather than having a Wulff morphology.”

Reference:

1. Chen, Y. *et al.* Kinetically controlled ammonia vapor diffusion synthesis of a Zn (ii) MOF and its H₂O/NH₃ adsorption properties. *J. Mater. Chem. A* 4, 10345-10351 (2016).
2. Smaldone, R. A. *et al.* Metal-organic frameworks from edible natural products. *Angew. Chem. Int. Ed* 49, 8630-8634 (2010).

Comment 2: Pore sizes and surface area are pivotal parameters for MOF materials. Given the nearly identical atom-pair distance and structural features of amorphous MOFs compared to crystalline MOFs (Figure 2), how are the pore sizes and surface areas of amorphous MOFs in this work? Additionally, it would be beneficial to explore whether their sizes and surface areas change with increased crystallinity under heat treatments (Figure S9).

Reply: According to your suggestions, we have tested the specific surface area and pore volume properties of amorphous MOFs. In general, the amorphous frameworks exhibit low specific surface area due to the collapse of intrinsic ordering framework (Figure S83-S85). With increasing crystallinity under heat treatments, the pore volume and the gas adsorption capacity of the material also increase. The related figures and contents are added as follows:

“The ordered pore structure and large specific surface area are characteristic features of MOFs. In this work, due to the collapse of the intrinsic ordered framework, the amorphous frameworks exhibit low specific surface area (Figure S83-S85). However, as colloidal particles transition from the amorphous to crystalline state, the pore volumes increase as well as the gas adsorption properties (Figure S85).”

Figure S83. The N₂ adsorption-desorption isotherm of a-MOF-74 (Fe) and a-Al-BTC sphere.

Figure S84. The CO₂ adsorption-desorption isotherm collected at 273 K of a-ZIF-zni and its corresponding crystalline product under heat treatment (90 °C).

Figure S85. The CO₂ adsorption-desorption isotherm collected at 298 K of a-ZIF-7 and its corresponding crystalline product under heat treatment (90 °C). The crystalline ZIF-7 exhibits a typical gate-open behavior in CO₂ adsorption-desorption.

	Maximum pore volume ($p/p^{\circ} = 0.016$)
Amorphous ZIF-7	0.011 cm ³ /g
Crystalline ZIF-7	0.089 cm ³ /g
Amorphous ZIF-zni	0.024 cm ³ /g
Crystalline ZIF-zni	0.033 cm ³ /g

Table S1. The maximum pore volume calculated based on CO₂ sorption.

Comment 3: Pre-surface modifications are typically required for growing MOFs on particles due to their weak interactions (ACS Nano 2015, 9, 7, 6951–6960; Nat. Chem., 2012, 4, 310–316). This work demonstrated an interesting, versatile growing of amorphous MOFs on diverse particles and claimed that the slow rate of TEA diffusion induces the MOF precursors to form heterogeneous nucleation on the surface of nanoparticles instead of homogeneous nucleation. Does the pH of the precursor solution govern the heterogeneous or homogeneous nucleation? More deep investigations are highly recommended, and understanding the dominant interactions between amorphous MOFs and representative core particles would also enhance the fundamental insights provided.

Reply: According to your suggestion, we have carefully investigated effect of pH on this synthetic method. By varying the pH of the precursors, the formation of Fe₂O₃@aZIF-7 were investigated. To eliminate the potential influence of surfactants, two different morphologies of surfactant-free Fe₂O₃ particles were selected. The results indicate that pH of precursor have significant effect on heterogeneous or homogeneous nucleation (**Figure S74-S76**), which demonstrate that the slow rate of TEA diffusion in this method plays a key role in inducing heterogeneous nucleation. The related contents and figures are added as follows:

“To further gain insight into the formation process of the core-shell structure, the effect of the pH of the precursors is carefully investigated and two surfactant-free Fe₂O₃ particles are selected as core-NPs. The direct addition of TEA into the precursor solution is applied to adjust the pH. In **Figure S74 and S75**, without any additional TEA added prior to the reaction (pH=7.45), uniform amorphous ZIF-7 forms around Fe₂O₃ particles, and no individual aZIF-7 colloids can be found. This indicates that heterogeneous nucleation dominates during this process. When the pH of the precursor solution reaches 9.35 (with the addition of 2.5 μl TEA), a thin layer of ZIF-7 deposits around the Fe₂O₃, but also a large number of individual aZIF-7 particles also emerge. A higher pH of the precursor solution accelerates the deprotonation of the ligands, resulting in simultaneous

occurrences of homogeneous and heterogeneous nucleation. When the pH increases even further, especially up to 9.90, homogeneous nucleation predominates, with hardly any aZIF forming shell structures around the Fe_2O_3 . These results demonstrate that the slow rate of TEA diffusion in this method play a key role in inducing heterogeneous nucleation (Figure S76).”

Figure S74. TEM images of Fe_2O_3 ellipsoids@aZIF-7 obtained by varying the pH of the precursor solution.

Fe₂O₃ nanoparticles:

Figure S75. TEM images of Fe₂O₃ NPs@aZIF-7 obtained by varying the pH of the precursor solution.

Figure S76. (a) The plot of the addition of TEA versus pH of the precursor solution. (b) The schematic illustration of the pH effect on the types of nucleation.

Comment 4: Is this amorphous coating technology also applied to planar substrates?

Reply: Expanding the technology to amorphous MOFs coatings onto thin films is crucial and is also one of our future research directions. Based on your suggestions, we selected copper plates as planar substrates for depositing amorphous MOFs. In **Figure S77**, aMOF-74 (Fe) was successfully deposited onto Cu plates. In our subsequent work, we will optimize the deposition conditions and conduct detailed research on the application of amorphous MOFs in thin film coatings.

Figure S77. (a) The optical and SEM images of pure Cu plate before. (b) The optical and SEM images of Cu plate after aMOF-74(Fe) deposition. (c) The Raman spectra of a-MOF-74(Fe) powder and Cu plate after aMOF-74(Fe) deposition. These SEM images and Raman spectra indicate the a-MOF-74(Fe) was successfully deposited on Cu plate.

Minor Comments:

Comment 1: Abstract: The authors claimed that the current challenge is to "achieving controllable synthesis," which is too broad and unclear. It is better to refine this central sentence to highlight the impact and novelty of the current work more clearly.

Reply: Thanks for your comment. According to your suggestion, we have revised the abstract to highlight our current work. The related sentence is listed below:

“It is important to note that amorphous coordination polymers and MOFs have demonstrated unique performances in energy storage and conversion, and drug delivery applications. However, achieving the synthesis of monodisperse colloidal particles and uniform core-shell structures has remained a challenge.”

Comment 2: Introduction, Page 2: The authors mentioned that "Recently, the Stöber method has been successfully extended to other materials including TiO₂ and resorcinol-formaldehyde (RF), 13...", in which both examples were published over 10 years ago. It would be great to highlight more recent advancements in new material composition design based on the Stöber method.

Reply: According to your suggestion, we have added other cases published in recent years. The related references are listed below:

14. Wei, J. et al. Sol-gel synthesis of metal-phenolic coordination spheres and their derived carbon composites. *Angew. Chem. Int. Ed.* **130**, 9986-9991 (2018).

15. Wang, G., Qin, J., Zhao, Y. & Wei, J. Nanoporous carbon spheres derived from metal-phenolic coordination polymers for supercapacitor and biosensor. *J. Colloid Interface Sci.* **544**, 241-248 (2019).

Comment 3: Figure 6b: Amorphous MOF coatings retain the similar shapes of the various substrates in Figures 5 and 6 except the b-FeOOH-1-based template (Figure 6B). Please provide a detailed explanation for clarity.

Reply: According to your suggestion, more detailed explanations are added as follows: Due to the higher surface energy at intersection points (**Figure S79**), the growth rate of aMOFs is faster, leading to the formation of nearly square core-shell structures. After a controllable etching process, cubic-like yolk-shell structures can be obtained.

Figure S79. Schematic illustration of the formation of yolk-shell structures of X-shape β -FeOOH@void@aZIF-zni. Due to the higher surface energy at intersection points, the growth rate of aMOFs is faster, leading to the formation of nearly square core-shell structures. After a controllable etching process, cubic-like yolk-shell structures can be obtained.

Comment 4: Many scale bars in microscopy images (e.g., Figure S2) are not easily visualized. Increase font size and bar thickness for better visibility, if possible.

Reply: Thanks for your suggestion, the related images are revised and some of them are selected as follows:

Figure 2. Structural characterizations of aZIF-7 spheres.

Figure S2. TEM images of amorphous ZIF-7 obtained by adding directly TEA to the mother solution.

Reviewer 3:

General comments: This manuscript introduced a general synthesis route to amorphous MOFs using a vapor diffusion method, which allows for the precise control of growth kinetics. The authors successfully synthesized 22 different amorphous MOF colloids by selecting 11 metal ions and 17 organic ligands. Additionally, they demonstrated the versatility of their amorphous coating technology by synthesizing over 100 new core-shell composites from 19 different amorphous MOF shells and 30 different core-NPs. This work significantly enriches the Stöber method and provides a platform for systematically designing colloids exhibiting varying levels of functionality and complexity. Overall, this is an interesting and valuable contribution to the field of MOF synthesis and their potential applications in energy storage, conversion, and drug delivery. Therefore, I recommend acceptance of the manuscript after the following issues have been addressed.

Reply: Thank you for your kind support. According to your suggestions, we added related data and discussions to improve this manuscript. Please find the point to point responses below.

Comment 1: Is it possible to replace the triethylamine with commonly used ammonia solution in the Stöber method as the base? Perhaps this approach could reduce costs.

Reply: Based on your suggestions, we selected Al-BTC as an example. Amorphous Al-BTC spheres were successfully synthesized via using ammonia solution as the source of base (**Figure S65**). The related figure is added as follows:

Figure S65. SEM images of a-Al-BTC spheres with different magnifications using an ammonia solution.

Comment 2: The pore structure is one of the key characteristics of MOF materials. Does this amorphous MOF material still retain its pore structure? The author can characterize one of the samples to help readers better understand the properties of the material.

Reply: Reviewer 2 also raised similar questions. Since the reply is the same, please refer to the answer to comment 2 of reviewer 2.

Comment 3: Figure 1c, some shape models do not conform to organic ligands, please confirm it.

Reply: Based on your comments, we have revised this figure to make it more clear :

Comment 4: Page 5 "As the temperature increases, the crystallinity also increases as demonstrated by the sharpening of the reflections in the PXRD pattern (Figure S9)". The author has studied that increasing temperature can improve crystallinity, but has the morphology changed during this process? It is recommended to characterize whether the sample with increased crystallinity still maintains its original morphology.

Reply: Based on your comments, we have tracked the transformation of morphology during heat treatment. As shown in **Figure S13**, the amorphous ZIF-7 spheres partially dissolve, reducing in size and the surface becomes rough, while new micron-sized crystalline ZIF-7 particles emerge. These results indicate that the amorphous colloidal particles underwent a partial dissolution-recrystallization process during crystallization. The related figures are added as follows:

Figure S13. SEM images of a-ZIF-7 spheres incubated at temperatures of 50°C for 24h with different magnifications. The obtained sample is a mixture of ZIF-7 spheres and micron-sized ZIF-7 particles.

Comment 5: Some writing errors need to be checked, such as line 3, page 13, "activate"

Reply: Thank you for your kind reminder. Based on your comments, we have carefully checked our manuscript and revised the related contents.

Reviewer 4:

General comments: In this communication, Zhang, Pinna et al. report on a simple strategy that mimics the Stöber method to synthesize diverse amorphous MOFs (aMOFs) spheres and aMOFs based core shell particles. The strategy is based on a TEA-based gas diffusion method to grow amorphous MOF colloids, including core-shell particles. The authors successfully synthesized a large number of new colloids using this original approach.

Personally, I must express my appreciation for this study and the innovative approach proposed by the authors. The idea is original and offers a general method to synthesize various amorphous

colloids. The study is well-organized, and the conclusions are well-supported by the data, primarily from microscopy but also incorporating some PDF measurements. I highly recommend its publication in Nature Communications.

Reply: Thanks for your kind support and positive comments. According to your suggestions, we have added more data and discussion to strength this manuscript. Please find the point to point responses below.

Comment 1: However, I do have a few questions regarding the potential applications of amorphous MOFs and how the authors plan to utilize this method in various applications: Given that the materials are amorphous and lack porosity and order, what criteria guide the selection of a specific MOF?

Reply: This work reports on the controllable synthesis of colloids of amorphous coordination polymers and MOFs. Our next step is to plan the exploration of potential applications for these amorphous colloids. The following aspects will be some criteria for selecting amorphous MOFs:

- 1) The characteristics of corresponding crystalline MOFs: Although lacking porosity and a long-range ordered structure, amorphous colloids still exhibit the typical chemical or physical properties of their crystalline phase. For example, in this work, amorphous Cr-NH₂-BDC spheres display fluorescence properties identical to crystalline phase (**Figure S61**).

- 2) The properties of metal or metal clusters nodes: For many reactions, open metal or metal cluster sites within MOFs exhibit high catalytic activity. More importantly, there are numerous

unsaturated metal or metal cluster sites distributed within the amorphous framework, typically displaying higher catalytic activity. Therefore, selecting the appropriate metal nodes allows for the design of specific catalysts.

- 3) As a template for synthesizing other materials: In addition to the direct carbonization of aMOFs to carbon or metal oxide, amorphous MOF-based core-shell colloids can be used as templates to construct corresponding crystalline MOF-based composites (**Figure S86**). Therefore, the intrinsic porosity, topological structure, and other characteristics of corresponding crystalline MOFs are also important factors to consider.

Figure S86. The TEM images of corresponding β -FeOOH/crystalline ZIF-7 composites under heat treatment (50 °C).

We have added the related discussion as follows:

“The potential applications of these amorphous colloids may arise from following aspects: 1) The characteristics of the corresponding crystalline MOFs is an important criterion to guide the selection. Although lacking porosity and a long-range ordered structure, amorphous colloids still exhibit the typical chemical or physical properties of the crystalline phase. 2) Abundant unsaturated metal sites within the amorphous framework exhibit robust catalytic activity, endowing promising potential for diverse catalytic systems. 3) Amorphous core-shell structures can be carefully converted into crystalline MOFs composites (**Figure S86**). This provides a new technological pathway for synthesizing crystalline MOF composites.”

Comment 2: In this case, can we define the shell as a MOF, or is it more accurately described as a coordination polymer?

Reply: Thanks for your comments. In this case, we have synthesized many types of amorphous colloids. Some of them such as a-ZIF-7 and a-MOF-74 (Fe) were synthetically investigated via using PDF, TEM, PXRD and gas-sorption. The PDF indicates nearly identical local structures and heat treatment successfully induced the conversion from amorphous to crystalline phase. These colloids could be defined as MOFs. But, this work involves the synthesis of 24 types of amorphous colloidal particles. Given the precious and limited synchrotron time for tests like PDF, it's challenging for us to conduct systematic testing on each one. I fully understand your concern about this. Therefore, in the future (this year and next year), we will apply for more synchrotron time to thoroughly characterize the local structure of amorphous colloids and their gas adsorption properties. Based on your suggestion, to make the description of this work more accurate, we will change amorphous MOFs to amorphous MOFs and coordination polymers in the manuscript. For example: “Stöber Method to Amorphous Metal Organic Frameworks and Coordination Polymers”

Comment 3: Is it possible to achieve similar results with any ligand or metal?

Reply: In this case, twenty-two different amorphous MOF or CPs colloids were successfully synthesized by selecting 11 metal ions and 17 organic ligands. To further demonstrate the generality of this method, two more CP colloids have been synthesized in this round (**Figure S68 and S69**). But, this method also has its limitations. For instance, the ligands must be soluble in alcohols at room temperature. This method is currently not applicable for ligands with low solubility. The related contents and figures are added as follows:

“It is noteworthy that this method also has its limitations. This strategy is currently not applicable for ligands with low solubility in alcohols at room temperature.”

Figure S68. (a) Schematic illustration of the preparation of amorphous Al-NH₂-BDC using the TEA diffusion method. (b) TEM images of a-Al-NH₂-BDC colloids. (c) TEM images of Mn₂O₃@a-Al-NH₂-BDC. (d) PXRD pattern of a-Al-NH₂-BDC colloids.

Figure S69. (a) Schematic illustration of the preparation of amorphous Er-PYDC using the TEA diffusion method. (b) TEM images of a- Er-PYDC colloids. (c) PXRD pattern of a-Er-PYDC colloids.

Comment 4: Are there any differences in the mechanical stability of the shell depending on the ligand or metal used?

Reply: Mechanical properties are crucial characteristics of MOFs, and there are some review articles describing factors influencing the mechanical stability of MOFs. Ligand and metal used indeed play a vital role in mechanical properties of MOFs. Therefore, in our system, different amorphous colloidal particles may exhibit some differences in mechanical stability. However, due to the temporary lack of conditions to experimentally test the mechanical stability of amorphous colloidal particles, in the future, we will seek simulated calculations and collaborate with other groups to study the mechanical stability of different amorphous colloids from experiment to calculation.

Comment 5: Additionally, some long-range SEM images would be beneficial to support the homogeneity of each sample.

Reply: According to your suggestions, we added the SEM images in this manuscript. Some of amorphous spheres are listed below:

Figure S4. SEM images of amorphous ZIF-7 with different magnifications obtained by TEA vapor diffusion method.

Amorphous ZIF-zni sphere

Figure S20. SEM images of amorphous ZIF-zni with different magnifications.

Amorphous ZIF-9 sphere

Figure S16. SEM images of amorphous ZIF-9 with different magnifications.

Figure S65. SEM images of amorphous Al-BTC with different magnifications.

Comment 6: These questions are raised in the spirit of enhancing the clarity and completeness of the study, and I look forward to the authors' insights on these aspects.

Reply: Thanks for your suggestions, we have revised the manuscript and added more data and discussion to improve this study.

REVIEWERS' COMMENTS

Reviewer #1 (Remarks to the Author):

The authors performed a large amount of additional experimentation, which strengthens the argument of this paper. The novelty and versatility of the proposed synthetic method is clear and deserves publication. The alphabet of the 14th molecule in Figure 3 is incorrect.

Reviewer #3 (Remarks to the Author):

The authors have addressed all my comments. Thus, I'd like to recommend its publication in the current form.

Reviewer #4 (Remarks to the Author):

I am pleased with the efforts made by the authors to address my concerns. I now recommend the publication of the manuscript in Nature Communications.

RESPONSE TO REVIEWERS' COMMENTS

Reviewer 1:

General comments: The authors performed a large amount of additional experimentation, which strengthens the argument of this paper. The novelty and versatility of the proposed synthetic method is clear and deserves publication. The alphabet of the 14th molecule in Figure 3 is incorrect.

Reply: Thanks for your kind support and reminder, we have corrected the 14th molecule in Figure 3

Reviewer 3

General comments: The authors have addressed all my comments. Thus, I'd like to recommend its publication in the current form.

Reply: Many thanks for your kind support.

Reviewer 4:

General comments: I am pleased with the efforts made by the authors to address my concerns. I now recommend the publication of the manuscript in Nature Communications.

Reply: Thank you for your kind support.